# Conservation Agriculture Saves Irrigation Water in the Dry Monsoon Phase in the Ethiopian Highlands

**Sisay A. Belay** [1,*], **Petra Schmitter** [2], **Abeyou W. Worqlul** [3], **Tammo S. Steenhuis** [1,4], **Manuel R. Reyes** [5] **and Seifu A. Tilahun** [1]

1   Faculty of Civil and Water Resources Engineering, Bahir Dar University, Bahir Dar 26, Ethiopia; tss1@cornell.edu (T.S.S.); satadm86@gmail.com (S.A.T.)
2   International Water Management Institute, Yangon 11081, Myanmar; P.Schmitter@cgiar.org
3   Blackland Research Center, Texas A&M AgriLife Research, Temple, TX 76502, USA; aworqlul@brc.tammus.edu
4   Department of Biological and Environmental Engineering, Cornell University, Ithaca, NY 14850, USA
5   Sustainable Intensification Innovation Lab, Kansas State University, Manhattan, KS 66506, USA; mannyreyes@ksu.edu
*   Correspondence: sisayasress@gmail.com

**Abstract:** Water resources in sub-Saharan Africa are more overstressed than in many other regions of the world. Experiments on commercial farms have shown that conservation agriculture (CA) can save water and improve the soil. Nevertheless, its benefits on smallholder irrigated farms have not been adequately investigated, particularly in dry monsoon phase in the Ethiopian highlands. We investigated the effect of conservation agriculture (grass mulch cover and no-tillage) on water-saving on smallholder farms in the Ethiopian highlands. Irrigated onion and garlic were grown on local farms. Two main factors were considered: the first factor was conservation agriculture versus conventional tillage, and the second factor was irrigation scheduling using reference evapotranspiration (ETo) versus irrigation scheduling managed by farmers. Results showed that for both onion and garlic, the yield and irrigation water use efficiency (IWUE) was over 40% greater for CA than conventional tillage (CT). The soil moisture after irrigation was higher in CA compared with CT treatment while CA used 49 mm less irrigation water. In addition, we found that ETo-based irrigation was superior to the farmers' irrigation practices for both crops. IWUE was lower in farmers irrigation practices due to lower onion and garlic yield responses to overirrigation and greater water application variability.

**Keywords:** conservation agriculture; conventional tillage; irrigation scheduling; farmers practice; irrigation water use efficiency

## 1. Introduction

Water resources in sub-Saharan Africa are limited and overstressed more than in many other regions of the world. Farmers grow one or two rain-fed crops per year, nevertheless, production is not sufficient to feed the current population. As a result, irrigation remains important to meet the needs of the people by increasing productivity. However, when more irrigation water uses are coupled with limited water availability, irrigation expansion in dry monsoon phases becomes a challenge [1–3]. Irrigation in combination with conservation agriculture has been used to save water and reduce offsite transport of fertilizer thereby increasing crop and water productivity. Scholars confirmed that one way to increase irrigated production is to use the available water more efficiently through the combined application of different irrigation and conservation agriculture (CA) practices [4,5].

Conservation agriculture involves maximum ground cover, minimum tillage, and the use of proper crop rotation [6].

Recent studies on large commercial agricultural areas found that conservation agriculture increased water productivity [7] and promoted soil health [8] and sustained agricultural resources [9–11], without compromising the crop yield. Few studies revealed that smallholder farmers can benefit more by combining conservation agriculture and irrigation practices. For example, authors in References [12] and [13] reported that irrigated cereals used less water under the combined or separate use of no-tillage, mulching, and crop rotation (elements of CA). However, in experiments using components of CA separately, scholars had often achieved less water-saving compared to the complete CA system [12]. Apart from high water-saving, the combined use of all components of conservation agriculture has shown to increase yield by about 30% [10]. Saving water is especially important in water-limited areas [14] and will allow increasing the irrigated acreage [15] and more smallholders to participate in the irrigation sector [6,11,16,17].

Only a few studies have been carried out on fields of smallholder farmers in sub-Saharan Africa, including Ethiopia [4,18–21]. Although these initial efforts were promising to increasing sustainable crop and water productivity [12], more research is needed in conservation agriculture-based irrigation practices in the dry phase in the Ethiopian highlands. The objective of this study is therefore to explore the combined impact of conservation agriculture and irrigation water management practices on water-saving, soil water dynamics, and related variables in the Ethiopian highlands.

## 2. Materials and Methods

### 2.1. Site Description and Experiment Features

The study area is located in Dengeshita experimental site in the headwaters of Blue Nile in the Northern Ethiopian highlands (around 11.32 N and 36.85 E at an altitude of 2042 m), 80 km south of Bahir Dar. The average rainfall during the main phase (June to September) is 1300 mm and during the dry phase (October to May) is 360 mm.

A total of 34 plots of 10 m by 10 m were established on farms to conduct an investigation on irrigated conservation agriculture (CA) during the dry phase from October 2016 to March 2018. Using random selection, 17 plots were assigned for conservation agriculture and 17 for conventional tillage (CT). The plots were selected based on the availability of a productive shallow well adjacent to the irrigable land, and farmers' willingness to participate. Onion and garlic were grown during the dry phase while hot pepper was grown using supplementary rain and is not considered here.

The plots have slopes ranging from two to five percent. The texture of the top 30 cm soil was a loam soil and inter-plot variation was insignificant using analysis of variance (Table 1). The texture of the 30–60 cm soil layer was generally a clay loam and, in some plots, the soil texture consisted of sandy loam. The soil was slightly acidic with a pH level of 6. Field capacity, permanent wilting point, bulk density, total nitrogen, available phosphorus, and available potassium in the top 30 cm were 0.31 $cm^3$ $cm^{-3}$, 0. 22 $cm^3$ $cm^{-3}$, 1.1 g $cm^{-3}$, 0.93 g $kg^{-1}$, 9.57 mg $kg^{-1}$, and 191 mg $kg^{-1}$, respectively.

Rainfall was recorded manually each morning at 6 A.M using a simple rain gauge installed near the experimental site (Figure 1). The remainder climate data used for calculating the reference evapotranspiration (ETo) with the FAO Penman–Monteith equation [22] were obtained from Dangila weather station for the period of 1995–2016. We excluded the years 1998–2000 from the period because of the large number of missing data. The climate data processed for the purpose include temperature (maximum and minimum), relative humidity, sunshine hours, and wind speed.

**Table 1.** Mean and standard deviation of physical and chemical properties of the soil from samples collected in 30 plots and at two depths of the experimental plots.

| Soil Parameter | Soil Depth | |
|---|---|---|
| | 0–30 cm | 30–60 cm |
| pH (H$_2$O) 1:2.5 | 6.0 ± 0.7 | 5.7 ± 0.7 |
| Cation Exchange Capacity CEC, cmol kg$^{-1}$ | 25.0 ± 4.7 | 24.0 ± 4.7 |
| Available phosphorus P, mg kg$^{-1}$ | 20.0 ± 14.1 | 6.9 ± 3.0 |
| Available potassium K, g kg$^{-1}$ | 1.0 ± 0.6 | 0.7 ± 0.4 |
| Total Nitrogen, TN, g kg$^{-1}$ | 0.2 ± 0.1 | 0.2 ± 0.1 |
| Field Capacity FC, cm$^3$ cm$^{-3}$ | 31.0 ± 3.5 | 28.0 ± 1.4 |
| Permanent wilting point PWP, cm$^3$ cm$^{-3}$ | 22.0 ± 4.2 | 21.5 ± 2.3 |
| Clay, g kg$^{-1}$ | 39.0 ± 18.0 | 16.3 ± 4.4 |
| Silt, g kg$^{-1}$ | 25.0 ± 4.9 | 23.3 ± 3.1 |
| Sand, g kg$^{-1}$ | 36.0 ± 19.0 | 60.3 ± 6.1 |
| Bulk Density, g cm$^{-3}$ | 1.1 ± 0.1 | 1.1 ± 0.2 |

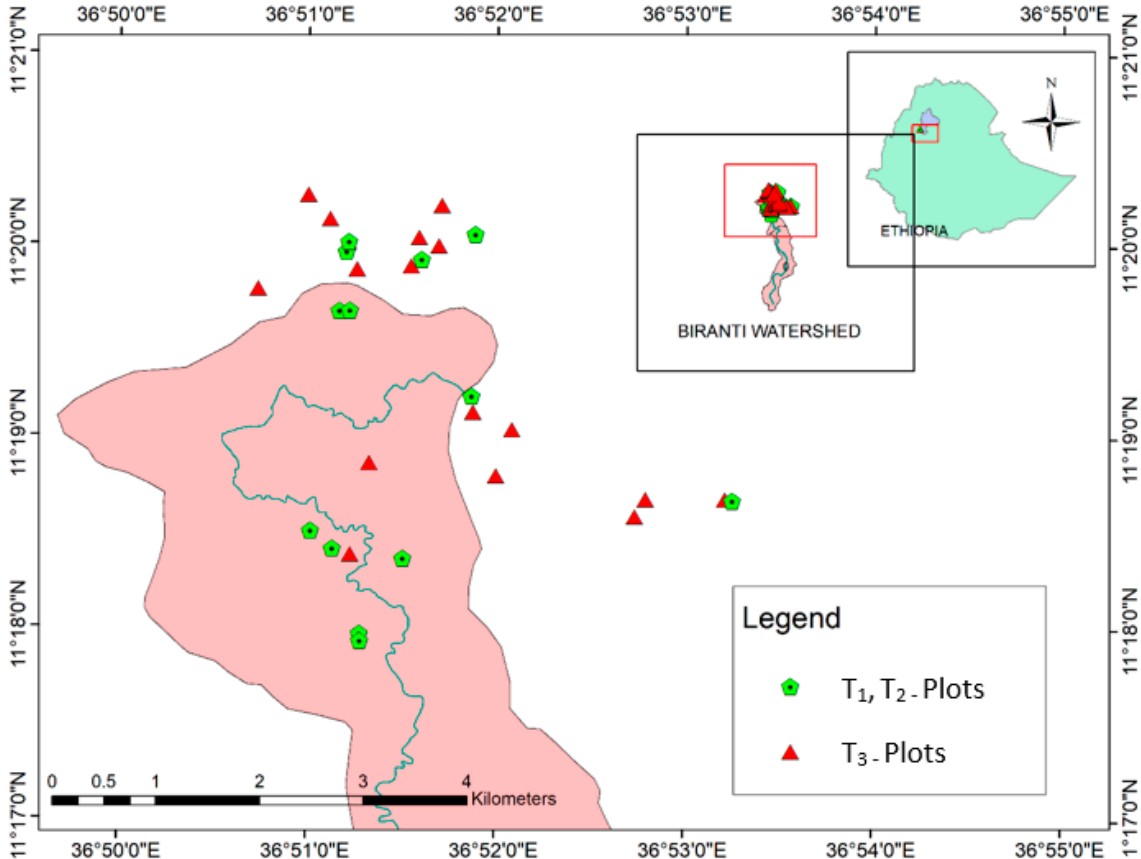

**Figure 1.** The location and geographical distribution of on-farm experimental plots in Dengeshita kebele (administrative unit smaller than district), in Northwestern Ethiopia, the state of Amhara. Some of the plots are within the Biranti watershed and the rest outside the watershed. The plots are located near the farmers' homes (Source: Ethiopian Mapping Agency).

Crop water use (ETc) was determined by multiplying ETo by the crop coefficient [23] for initial, development, mid-season, and end stages (Table 1). Irrigation water to be applied to onion and garlic was determined at an allowable constant soil moisture depletion fraction (*f* = 0.25) of the total available soil water (TAW), where TAW was determined from the permanent wilting point, field capacity, root depth, and bulk density variables. The depth of water applied during each irrigation event was

the net irrigation requirement between irrigation events, plus that needed for inefficiencies in the irrigation system. In this experiment, considering application losses, an irrigation efficiency of 80% was assumed and added to each plot.

## 2.2. Experimental Design

Two main factors were considered: the first factor was conservation agriculture versus conventional tillage practices, and the second factor was irrigation scheduling using reference evapotranspiration (ETo) versus irrigation scheduling managed by farmers' practices. Conservation agriculture consists of no-tillage and application of grass mulch at the rate of 2 t ha$^{-1}$, while conventional tillage is the current farmers' practice of 4–6 tills and without mulch cover. Irrigation water amount and scheduling managed by estimated reference evapotranspiration (ETo) here refer to the use of calculated crop water requirement (ETc) estimated from ETo. Accordingly, the treatments were:

$T_1$: conservation agriculture with irrigation water amount and scheduling managed by estimated evapotranspiration; $T_2$: conventional tillage with irrigation water amount and scheduling managed by estimated evapotranspiration and, $T_3$: conservation agriculture with irrigation water amount and scheduling managed by farmers' practices.

The three treatments were conducted with onion crop in 2016/2017 replicated 17 times on 17 on-farm plots (Figure 1), and with garlic in 2017/2018 replicated 14 times (Figure 1). In 2016/2017, treatments $T_1$ and $T_2$ received similar irrigation volume and scheduling practice while in 2017/2018, the two treatments were irrigated differently. Treatments $T_1$ and $T_2$ were on the same plot where half was for $T_1$ and half for $T_2$ with pair-t design. The amount of irrigation applied was measured by counting the number of known volume buckets (or watering cans) per each application.

On all plots, a similar rotation of onion, green pepper, and garlic was followed: onion was planted on 20/12/2016 and harvested on 25/3/2017. It was followed by green pepper from 1/5/2017 to 10/9/2017 and then garlic from 18/10/2017 to 26/2/2018. Since pepper was grown partially in rain and dry phase, it was excluded from this paper. The location of the plots distribution for each treatment is shown in Figure 1.

## 2.3. Crop Variety and Management Information

Adama Red Onion (Allium cepa L.) local variety in 2016/2017 and garlic (Allium Sativium L.) local variety in 2017/2018 were transplanted or planted respectively on 20/12/2016 and 18/10/2017, at a spacing of 20 cm between rows and between plants [24–26]. Onion seedlings were transplanted at the age of 50 days. Crop coefficients are shown in Table 2, and the management activities for growing onion and garlic vegetables are shown in Table 3.

**Table 2.** Crop stages, length of the growing period in days, and crop coefficients.

| Year | Crop Type | Crop Parameters | Growth Stages | | | |
|------|-----------|-----------------|---------|-------------|------------|-----|
| | | | Initial | Development | Mid-Season | End |
| 2017 | Onion | Length of growth (days) | 20 | 45 | 35 | 20 |
| | | Crop coefficient (Kc) | 0.7 | 0.7–1.05 | 1.05 | 0.7 |
| 2018 | Garlic | Length of growth (days) | 20 | 50 | 30 | 20 |
| | | Crop coefficient (Kc) | 0.7 | 0.7–0.95 | 0.95 | 0.7 |

**Table 3.** Experimental onion and garlic varieties, management activities, date of operation, and method of cultivation performed at the study site (2016 to 2018 years) over the growing seasons.

| Year | Crop | Activities | Date (Day/Month/Year) | Method |
|------|------|-----------|----------------------|--------|
| 2016/2017 | Adama Red Onion (Allium cepa L.) | Seedling | 2/11/2016 | Manual |
| | | Tillage * | 25/9/2016–30/3/2016 | Oxen and Manual |
| | | Transplanting | 20/12/2016 | Manual |
| | | Mulch application ** | 1/5/2017 | Manual |
| | | Irrigation | 20/12/2017–20/3/2017 | watering-Can |
| | | weeding/hoeing * | 20/1/2017, 29/2/2017, 16/3/2017 | Manual |
| | | Harvesting | 22/3/2018–25/3/2017 | Manual |
| 2017/2018 | Local garlic (Allium Sativium L.) | Tillage * | 9/10/2017–14/10/2017 | Oxen and Manual |
| | | Planting | 18/10/2017 | Manual |
| | | Mulch application ** | 27/10/2017 | Manual |
| | | Irrigation | 27/10/2017–26/1/2018 | watering-Can |
| | | weeding/hoeing * | 27/11/2017, 29/12/2017, 16/1/2018 | Manual |
| | | Harvesting | 26/2/2018 | Manual |

* = For conventional tillage treatment, tillage was practiced but no grass-mulch was used; ** = For conservation agriculture no-tillage and grass-mulch were practiced.

Each experimental plot was equally treated with urea fertilizer (46-0-0) at a rate of 200 kg ha$^{-1}$ and applied according to local management practices. Local weed seed free grass species were harvested and dried for mulching in order to prevent conservation agriculture plots from weed infestation. The grass mulch was applied to CA plots at the rate of 2 t ha$^{-1}$ in each experimental period. Crop phenological variables such as height and number of leaves were measured every 10 days by randomly selecting nine plants from each plot. The onion was harvested on 22–25 April 2017 while garlic was harvested on 18 February 2018.

*2.4. Soil Moisture Data*

In 2017/2018 in garlic, soil moisture at the top 20 cm depth was monitored using time domain reflectometry (TDR) probes (TDR 200 Spectrum Technology Inc.). The TDR was not installed type. Rather, two agricultural extension agents were trained to measure the soil moisture each time by inserting a pair of 20 cm length TDR rods into the soil. TDR measurement was conducted before and after an irrigation event (3 times a week) for only $T_1$ and $T_2$ treatments because our interest was to compare the effect of conservation agriculture ($T_1$) and conventional tillage ($T_2$) on soil moisture content. In addition to TDR measurement, the soil moisture over the top 10, 20, and 30 cm soil depth were monitored using gravimetric method once every 10 days. TDR probes were calibrated using gravimetric soil moisture determination technique to increase the data quality. Irrigation was ceased 2 weeks before harvest to prevent both onion and garlic tubers from rotting and sprouting [27]. Irrigation water use efficiency (IWUE) in kg m$^{-3}$ was estimated by dividing fresh yield of onion or garlic by the volume of irrigation water applied to grow each of the vegetables.

*2.5. Data Analysis*

All data are presented with arithmetic means and were statistically analyzed using analysis of variance (ANOVA) after checking the normality using W/S and Jarque–Berra methods [28]. All the results shown in tables and figures are means of treatment plots. Mean values were compared for any significant differences using the least significant difference (LSD) method. LSD was calculated from data, where the differences among means were tested at $\alpha = 0.05$.

## 3. Results and Discussion

### 3.1. Irrigation Water Applied

The total irrigation water applied to the onion crop was 520 mm for both $T_1$ and $T_2$. Irrigation in these treatments was managed by replacing the water lost in crop evapotranspiration (ETc) three times per week assuming an 80% irrigation efficiency (Table 4). The total irrigation water used in $T_3$ was 548. Irrigation water amount in this treatment was determined by farmers' practice. Water used in $T_3$ was much greater than $T_1$ or $T_2$ though the difference was insignificant ($P < 0.05$). Similarly, a significantly greater amount of water was used for garlic crop in the $T_3$ treatment (Table 4). In both crops, the total irrigation water applied to $T_1$ was the smallest while it was the highest for $T_3$. The total water applied for garlic was 14% and 45% less in $T_1$ compared to $T_2$ and $T_3$, respectively.

**Table 4.** Applied water (mm) to each growth stages of onion and garlic vegetables and the variations using analysis of variance ($\alpha = 0.05$) *.

| Treatment * | Crop Stages | | | | |
|---|---|---|---|---|---|
| | Initial | Development | Mid-Season | End | Total |
| Onion in 2016/2017 | | | | | |
| $T_1$ | 136 [a*] | 219 [a**] | 122 [a] | 42 [a] | 520 [a] |
| $T_2$ | 136 [a] | 219 [a] | 122 [a] | 42 [a] | 520 [a] |
| $T_3$ | 157 [b] | 213 [a] | 141 [b] | 36 [a] | 548 [a] |
| P-value | 0.04 | 0.80 | 0.09 | 0.50 | 0.40 |
| LSD$_{(\alpha = 0.05)}$ | 20.80 s | 35.4 ns | 23.0 ns | 19.2 ns | 66.8 ns |
| Garlic in 2017/2018 | | | | | |
| $T_1$ | 48 [a] | 120 [a] | 59 [a] | 33 [a] | 260 [a] |
| $T_2$ | 55 [ab] | 142 [ab] | 73 [ab] | 39 [ab] | 309 [ab] |
| $T_3$ | 70 [b] | 194 [c] | 86 [bc] | 50 [bc] | 420 [c] |
| P-value | 0.0025 | 0.0004 | 0.017 | 0.015 | 0.00095 |
| LSD$_{(\alpha = 0.05)}$ | 15.14 | 43.6 15.14 | 22.52 | 15.91 | 87.72 |

* Numbers followed by same letters under same heads in a column are statistically nonsignificant at $\alpha = 0.05$ significant level; $T_1$: conservation agriculture with irrigation water amount and scheduling managed by estimated evapotranspiration; $T_2$: conventional tillage with irrigation water amount and scheduling managed by estimated evapotranspiration and; $T_3$: conservation agriculture with irrigation water application managed by farmers' practices.

Irrigation water applied at the initial stage to onion was 136 mm in $T_1$ and 157 mm in $T_3$. However, irrigation water applied to the initial stage of garlic was, respectively, 48, 55, and 70 mm for $T_1$, $T_2$, and $T_3$ treatments. Due to the season of transplanting, onion received greater irrigation water application than the garlic crop. The onion was transplanted during a much drier month on 20/12/2016, while garlic was planted during a much wetter month on 10/10/2017 (Table 3). Correspondingly, the soil moisture after the end of the rainy season was higher in garlic production period, and hence less irrigation water was applied. Water applied at the initial stage of onion was 46% less in $T_1$ or $T_2$ compared with $T_3$ treatment. Similarly, there was 15% and 31% less water used for garlic in $T_1$ compared with $T_2$ and $T_3$ treatments, respectively. Less irrigation water was used for garlic in conservation agriculture ($T_1$) than conventional tillage ($T_2$) treatment. The reason was attributed to grass mulch cover and no-tillage practices in $T_1$ treatment. The depth of irrigation applied at initial stage of onion and the development stage of garlic was significantly ($P < 0.05$) higher in $T_3$ compared with $T_1$ and $T_2$ treatment. On the other hand, in similar conservation agriculture treatment ($T_1$ and $T_3$), farmers scheduling practice ($T_3$) used more water particularly at initial and mid-season stages for onion and at all stages for garlic compared to estimated evapotranspiration irrigation scheduling ($T_1$).

Irrigation interval in $T_1$ and $T_2$ was 2 days for the onion crop and 3 days for the garlic crop (Table 5). The recommended irrigation amount per application was similar in $T_1$ and $T_2$ for onion

crop. The irrigation interval under farmers' scheduling practices ($T_3$) varied between 1 and 4 days (Table 5). Greater amount of water was also applied per irrigation (Figure 2). The variability in farmers' irrigation practice ($T_3$) was primarily governed by labor availability, and therefore the depth of water applied per application was different (Figure 2).

**Table 5.** Irrigation interval, depth of water application, and the total number of irrigations practiced for onion and garlic production.

| Treatment | Irrigation Interval (days) | Irrigation Depth per Application (mm) | Number of Irrigations |
|---|---|---|---|
| **Onion—2016/2017** | | | |
| $T_1$ * | 2 | 5–8 | 80–70 |
| $T_2$ | 2 | 5–8 | 80–70 |
| $T_3$ | 1–4 | 4–10 | 90–60 |
| **Garlic—2017/2018** | | | |
| $T_1$ | 3 | 6–8 | 40 |
| $T_2$ | 3 | 8–10 | 40 |
| $T_3$ | 2–4 | 5–13 | 50 |

* $T_1$: conservation agriculture with irrigation water amount and scheduling managed by estimated evapotranspiration; $T_2$: conventional tillage with irrigation water amount and scheduling managed by estimated evapotranspiration and; $T_3$: conservation agriculture with irrigation water amount and scheduling managed by farmers' practices.

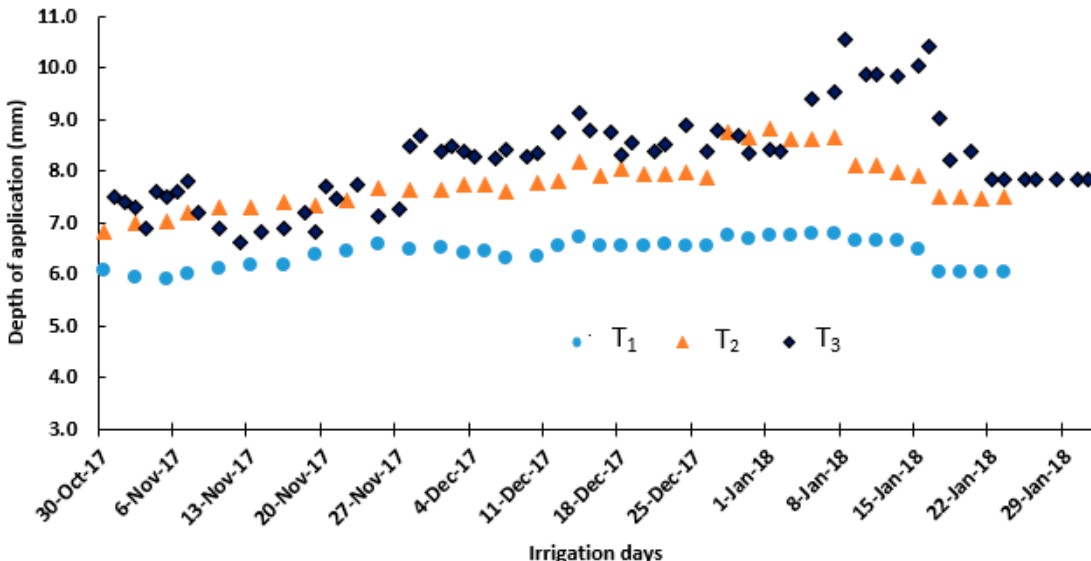

**Figure 2.** Average depth of irrigation per application for garlic crop since planting for the three treatments ($T_1$, $T_2$, and $T_3$). The treatments $T_1$, $T_2$, and $T_3$ are described in Table 4.

Cumulative irrigation water used in each treatment, cumulative estimated evapotranspiration (ETc) used in $T_1$ and $T_2$, and cumulative rainfall during the growing season of the two crops was depicted is in Figure 3. It shows that $T_3$ (farmers' practice) received a significantly greater amount of water at any stage compared with $T_1$ and $T_2$ treatments (estimated evapotranspiration-ETc). In both crops, the irrigation water used for the treatments was only slightly greater than the estimated ETc. Onion received less rainfall than garlic after transplanting (Figure 3a,b).

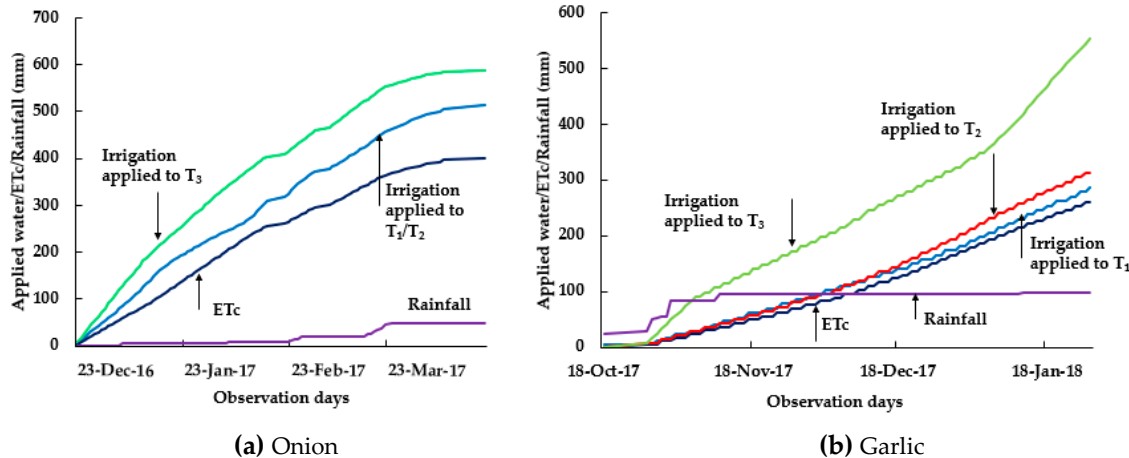

**(a)** Onion																																**(b)** Garlic

**Figure 3.** Cumulative estimated evapotranspiration (ETc), depth of irrigation application, and rainfall for (**a**) onion (2016/2017), and (**b**) garlic (2017/2018) vegetables; * $T_1$: conservation agriculture with irrigation water amount and scheduling managed by estimated evapotranspiration; $T_2$: conventional tillage with irrigation water amount and scheduling managed by estimated evapotranspiration and; $T_3$: conservation agriculture with irrigation water amount and scheduling managed by farmers' practices.

### 3.2. Soil Moisture Dynamics Responses

The soil moisture in $T_1$ and $T_2$ treatments was monitored using TDR probes for only the garlic crop period and is shown in Figure 4. It was not measured for the onion crop in 2016/2017. Soil moisture was measured only under $T_1$ and $T_2$ treatments because we wanted to compare conventional tillage ($T_2$) with conservation agricultural ($T_1$) practices. In Figure 4, soil moisture before and after irrigation is shown by dashed and solid blue lines under conservation agriculture treatment ($T_1$), while it is also shown by dashed and solid red lines under conventional tillage ($T_2$). Correspondingly, the available soil moisture after irrigation in $T_1$ is indicated by the area bounded by the blue lines and are shaded by vertical lines. Similarly, the area bounded by red lines and colored yellow represents the available soil moisture after irrigation in $T_2$. Region A indicates the soil moisture gained in $T_1$ over $T_2$ after irrigation, Region B is the common soil moisture for the two treatments after irrigation, and Region C indicates the soil moisture deficit under $T_2$ before irrigation. This difference (significant at $P < 0.05$) in soil moisture was attained in conservation agriculture ($T_1$) over conventional tillage ($T_2$) while it received 49 mm less applied water than conventional tillage treatment ($T_2$) due to reduced evaporation of the grass mulch cover [29].

In addition to the TDR measurements, we took gravimetric soil moisture contents at 10, 20, and 30 cm depth by taking soil samples seven times during garlic growing season for the $T_1$ and $T_2$ treatments (Figure 5). Figure 5 shows that the soil moisture (%) in treatment $T_1$ (solid line) was greater than $T_2$ (dashed line) during the garlic growing period. The greatest difference in soil moisture variation was observed in the surface 10 cm ($T_1$-10 and $T_2$-10) soil layer and the smallest in the lowest 30 cm ($T_1$-30 and $T_2$-30) soil layer (Figure 5). This shows that despite less irrigation water was applied to $T_1$, the soil moisture in $T_1$ was greater compared with $T_2$.

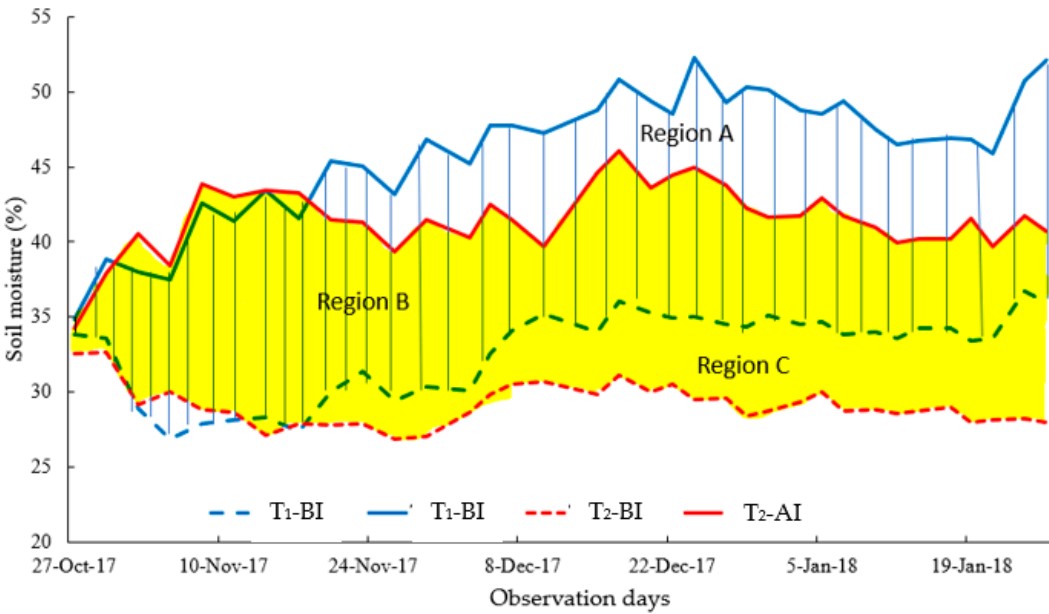

**Figure 4.** Soil moisture of $T_1$ and $T_2$ treatments for garlic crop measured at the top 20 cm soil layer before and after irrigation water application. Region A indicates the soil moisture gained in $T_1$ over $T_2$ after irrigation, Region B is common for the two treatments, and Region C indicates the soil moisture deficit of $T_2$ before irrigation. * $T_1$: conservation agriculture with irrigation water amount and scheduling managed by estimated evapotranspiration; $T_2$: conventional tillage with irrigation water amount and scheduling managed by estimated evapotranspiration. $T_1$-BI = soil moisture (%) before irrigation for ($T_1$) treatment; $T_1$-AI = soil moisture (%) after irrigation $T_1$ treatment. $T_2$-BI = soil moisture (%) before irrigation for ($T_2$) treatment; $T_2$-AI = soil moisture (%) after irrigation $T_2$ treatment.

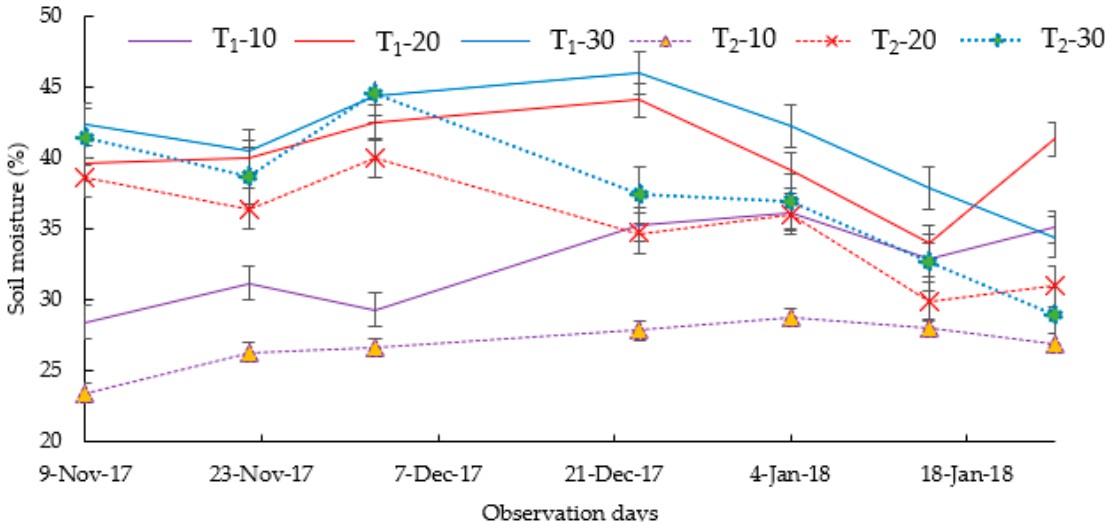

**Figure 5.** Soil moisture dynamics at 10 cm, 20 cm, and 30 cm soil depths for $T_1$ and $T_2$ treatments under garlic crop experiment. $T_1$-10, $T_1$-20, and $T_1$-30 indicates soil moisture measurements at 10 cm, 20 cm, and 30 cm depth for conservation agriculture ($T_1$); and $T_2$-10, $T_2$-20, and $T_2$-30 indicate soil moisture measurements at 10 cm, 20 cm, and 30 cm for conventional tillage ($T_2$), monitored once every 10 days after planting of garlic.

### 3.3. Yield and Productivity

The yield of onion and garlic was greater and statistically significant ($P < 0.05$) in conservation agriculture ($T_1$) compared with conventional tillage practices ($T_2$) (Table 6). The yield of onion was 24, 18, and 15 t ha$^{-1}$ respectively in $T_1$, $T_2$, and $T_3$ treatments (Table 6). A high yield of onion in $T_1$ is associated with improved soil moisture due to grass mulch cover (Figures 4 and 5). Moreover, it was observed that grass mulch cover prevented the emergence and regrowth of weeds. It, therefore, reduced the competition for water and nutrients.

**Table 6.** Average irrigation water applied yield, productivity, and irrigation water use efficiency (IWUE) values for each treatment. Significant and mean differences among treatments were performed using analysis of variance ($\alpha = 0.05$) and Tukey Least Significant Difference (LSD) method *.

| Treatments | Applied Water (mm) | Yield (kg) | Yield (t ha$^{-1}$) | IWUE (kg m$^{-3}$) |
|---|---|---|---|---|
| | | Onion 2016/2017 | | |
| $T_1$ | 520 [a] | 54.7 [a] | 24.3 [a] | 4.42 [a] |
| $T_2$ | 520 [a] | 40.1 [b] | 17.9 [b] | 3.24 [b] |
| $T_3$ | 548 [a] | 65.1 [c] | 14.9 [b] | 2.40 [b] |
| P-value | 0.4 | <0.01 | 0.12 | 0.00004 |
| LSD (0.05) | 66.8 | 8.5 | 3.4 | 0.77 |
| | | Garlic 2017/2018 | | |
| $T_1$ | 260 [a] | 15.2 [a] | 5.3 [a] | 1.9 [a] |
| $T_2$ | 309 [ab] | 11.0 [bc] | 3.8 [a] | 1.2 [bc] |
| $T_3$ | 420 [c] | 12.6 [ac] | 3.8 [a] | 1.3 [c] |
| P-value | 0.00095 | <0.01 | 0.187 | 0.006 |
| LSD (0.05) | 87.7 | 3.1 | 1.7 | 0.5 |

* Numbers followed by same letters under same heads in a column are statistically nonsignificant by LSD test at $P < 0.05$. $T_1$: conservation agriculture with irrigation water amount and scheduling managed by estimated evapotranspiration; $T_2$: conventional tillage with irrigation water amount and scheduling managed by estimated evapotranspiration and; $T_3$: conservation agriculture with irrigation water amount and scheduling managed by farmers' practices.

In the CA treatment, the yield in $T_3$ was lower compared with $T_1$. The reason for the yield reduction was related with suboptimal irrigation intervals for $T_3$ that caused either overwatering or underwatering (Table 5). Treatment $T_3$ received only slightly higher irrigation water application (548 mm) in onion production than $T_1$ and $T_2$ (520 mm), however, excess water was applied at the initial stage, and most of it was lost through percolation. In addition, we observed that the thick grass mulch cover made it difficult for the farmers to identify the soil wetness. Hence, they over- or under-irrigated their fields. Our results are in agreement with the findings of other experiments in Bangladesh [29] that irrigation water application affects crop growth by influencing the availability of water and nutrients, and therefore, it needs to be managed carefully [30]. The results were also consistent with the findings reported in Reference [31]. The recommendation of Reference [32] indicated that the water availability during vegetative development stage is directly linked with the most important stage to maximize canopy formation and yield. These results agree with onion yields reported in Reference [4] under on-station drip research conducted in the semi-arid region of Ethiopia.

Similarly, the yields of garlic were 5.3, 3.8, and 3.8 t ha$^{-1}$, respectively, for $T_1$, $T_2$, and $T_3$ treatments (Table 6). The reason for the significantly higher yield of garlic in $T_1$, compared with $T_2$, was similar to that discussed above for onion. In similar CA treatment, the yield in $T_3$ was lower compared with $T_1$. The reason for the garlic yield reduction in $T_3$ could be associated with much longer or shorter irrigation intervals that caused overwatering or underwatering (Table 5). In other words, the distribution of soil moisture in $T_3$ was not uniform and led to a decrease in yield. The results are in agreement with the findings of an experiment in Bangladesh [29]. Moreover, the result of Reference [33] agrees with this study. Water and other inputs interact with each other and their improper combination could reduce

yield as reported in Reference [34]. Shock [32] recommended that the vegetative development stage is the most important stage to maximize canopy formation and yield. Similar garlic yield results were also reported in Reference [29] under zero tillage and water hyacinth mulch combination. Adekpe [34] reported garlic yield result for Africa which is in harmony with this study. Under similar region of this study, onion yield was reported by scholars in Ethiopia [35–38]. All the results discussed earlier slightly vary due to many experimental factors. Slight variations were noted due to the type of experiment (on-station or on-farm), size of the experiment (smallholder or large commercial), crop intensity, and local management differences.

### 3.4. Crop Growth Dynamics and Responses

Greater bulb weights and higher crop height were achieved with conservation agriculture ($T_1$) than with conventional tillage practices ($T_2$) in both experimental years. In 2016/2017, onion bulb weight obtained from $T_2$ was smaller (30 to 60 g) in size than $T_1$ treatment (40 to 80 g). The bulb weight difference was related to higher soil moisture content (Figure 5). The grass mulch cover under CA solves water deficiency and adequately recharged the onion root zone as reported in Reference [39]. This also agrees with the findings in References [13] and [24]. In 2017/2018, greater garlic bulb weights were also obtained in $T_1$ (60 to 80 g) compared with $T_2$ (30 to 40 g) treatment. These results are in agreement with the findings in Reference [40] where larger garlic bulbs were obtained in water hyacinth mulch than non-mulch practices. The work in Reference [34] is also consistent with the results of this study.

The onion bulb height was highest in $T_1$ and lowest in $T_3$ treatment (Figure 6). Onion yield was directly proportional to the onion bulb height which is also in agreement with Reference [41]. Similarly, the garlic bulb height was the highest in $T_1$ and the lowest in $T_2$ (Figure 7). In both crops, the bulb height in $T_1$ was almost higher than $T_2$ and $T_3$ at any observation day. The variation between $T_1$ and $T_2$ treatments was also statistically significant ($p < 0.05$). This result is also consistent with the results reported in References [35–38].

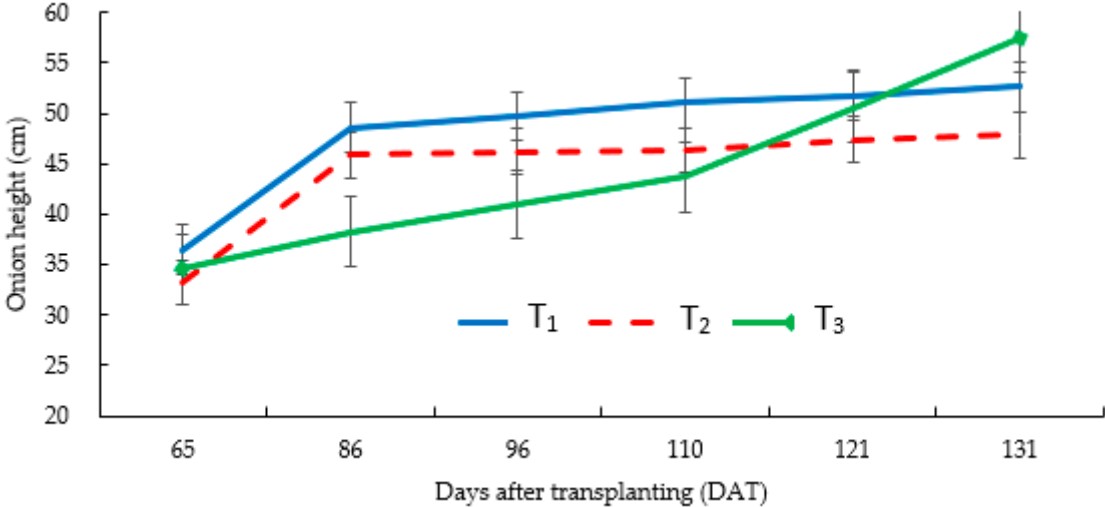

**Figure 6.** Onion height measured at 10-day intervals after transplanting and responses to conservation agriculture among treatments, * $T_1$: conservation agriculture with irrigation water amount and scheduling managed by estimated evapotranspiration; $T_2$: conventional tillage with irrigation water amount and scheduling managed by estimated evapotranspiration and; $T_3$: conservation agriculture with irrigation water amount and scheduling managed by farmers' practices.

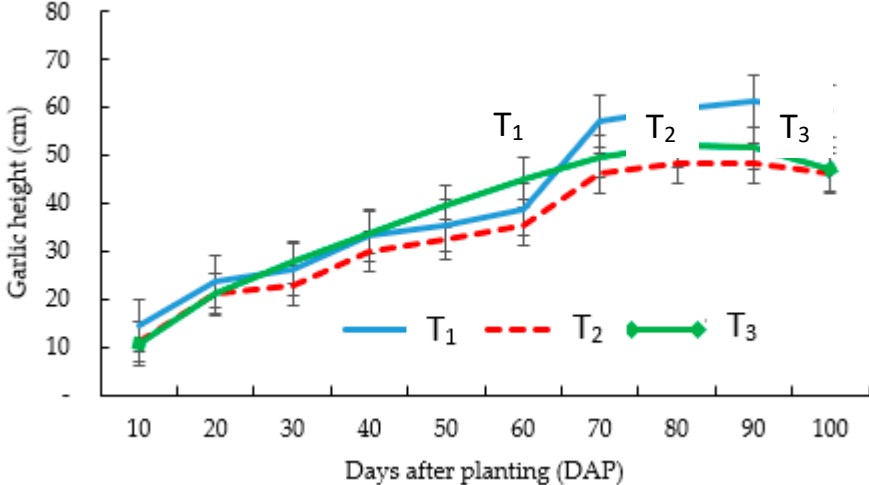

**Figure 7.** Garlic height measured at 10-day intervals after transplanting and responses to conservation agriculture among treatments, * $T_1$: conservation agriculture with irrigation water amount and scheduling managed by estimated evapotranspiration; $T_2$: conventional tillage with irrigation water amount and scheduling managed by estimated evapotranspiration and; $T_3$: conservation agriculture with irrigation water amount and scheduling managed by farmers' practices.

The difference in bulb weight and height could be associated with a conducive environment within the soil by CA practices. The grass mulch cover under CA kept the water needed by the crop consistent in time (Figures 4 and 5) and adequately recharged the onion root zone as reported in Reference [42]. This explanation agreed well with the findings in References [15] and [27]. These findings strengthen the role of conservation agriculture to solve sudden water stress for better yield of onion and garlic.

*3.5. Irrigation Water Use Efficiency (IWUE)*

Irrigation water use efficiency (IWUE) for onion and garlic vegetables was increased in conservation agriculture ($T_1$) compared to conventional tillage ($T_2$) (Table 6). IWUE of onion was 4.4, 3.2, and 2.4 kg $m^{-3}$, respectively, in $T_1$, $T_2$, and $T_3$ treatments. This shows that IWUE in $T_1$ treatment was 44% higher than $T_2$ and 76% higher than $T_3$ treatment. The difference in IWUE between $T_1$ and $T_2$ was also statistically significant at $\alpha = 0.05$ significant level with LSD = 0.77 (Table 6).

Similarly, IWUE of garlic was 1.9, 1.2, and 1.3 kg $m^{-3}$, respectively, in $T_1$, $T_2$, and $T_3$ treatments. It implies that IWUE in $T_1$ treatment was 57% higher than $T_2$ and 49% higher than $T_3$ treatment (Table 6). $T_1$ was significantly different ($p < 0.05$) from the other two treatments. Due to lower yield response to higher irrigation water application at initial stages, IWUE was the lowest in $T_3$ for onion while IWUE for garlic was the lowest in $T_2$. Al-Jamal [42] in New Mexico reported the IWUE of sprinkler and furrow irrigated onion experiment. However, the results are significantly higher than our results probably because of the high level of nutrient provision. IWUE results of this study are also consistent with the results reported under furrow irrigated experiments in Texas [43]. In harmony with our results, IWUE results were reported in surface drip on-station onion experiment in India [31]. Similar IWUE results were reported under a greenhouse pot experiment conducted in Turkey [44]. Moreover, IWUE was also reported under low head drip irrigated onion experiment in the semiarid region of Ethiopia [4]. A low-level IWUE was found in a drip-irrigated onion experiment in southern Ethiopia [45].

Generally, the IWUE results of this study under onion and garlic vegetables were comparable to other reported values in similar regions (Table 7). It showed that the IWUE shown in Table 7 were within the range of 2.2–17.5 kg $m^{-3}$ for onion, and in the range of 1.1–3.9 kg $m^{-3}$ for garlic depending on the differences in climate and fertilizer management. The yield results in this study lay within these ranges.

**Table 7.** Comparisons of experimental findings in applied irrigation water (mm), yield (t ha$^{-1}$), and IWUE (kg m$^{-3}$) under irrigated onion and garlic experimental studies.

| References | Location | Type of Experiment | Experimental Crop | Treatment Type | Irrigation Method | Applied Water (mm) | Yield (t ha$^{-1}$) | IWUE (kg m$^{-3}$) |
|---|---|---|---|---|---|---|---|---|
| [46] | Los Ebanos, Texas, USA | commercial | onion | irrigation methods | surface drip | 359 | 62.9 | 17.5 |
| [46] | Los Ebanos, Texas, USA | commercial | onion | irrigation methods | furrow | 677 | 28.7 | 4.2 |
| [47] | Arkansas, USA | commercial | onion | irrigation methods | furrow | 640 | 35.0 | 5.5 |
| [4] * | Sekota, Ethiopia | station | onion | irrigation scheduling | Drip | 278 | 25.0 | 9.0 |
| [27] | Abohar, Punjab, India | station | onion | deficit irrigation | Micro-sprinkler | 275 | 19.0 | 6.9 |
| [27] | Abohar, Punjab, India | station | onion | deficit irrigation | Micro-sprinkler | 467 | 36.0 | 7.7 |
| [31] | India | station | onion | deficit irrigation | Subsurface | 563 | 44.4 | 7.9 |
| [31] | India | station | onion | deficit irrigation | Subsurface drip | 328 | 28.1 | 8.6 |
| [46] | Los Ebanos, Texas, USA | station | onion | deficit irrigation | Subsurface drip | 389 | 42.0 | 10.8 |
| [46] | Los Ebanos, Texas, USA | station | onion | deficit irrigation | Subsurface drip | 292 | 39.0 | 13.4 |
| [44] | Turkey | GH pot [1] | onion | deficit irrigation | sprinkler | 190–680 | 4.4–27 | 2.2–5.6 |
| [45] | Jima, Ethiopia | On-farm | onion | Variety | Drip | 315 | 6.9 | 2.2 |
| [48] | California, USA | On-farm | Garlic | Irrigation Interval | 1 week | 350 | 21.3 | 6.1 |
| [48] | California, USA | On-farm | Garlic | irrigation interval | 1.5 week | 300 | 19.1 | 6.4 |
| [29] | Mymensingh, Bangladesh | station | Garlic | CA | drip | 446 | 7.8 | 1.7 |
| [29] | Mymensingh, Bangladesh | station | Garlic | CA | drip | 546 | 6.8 | 1.2 |
| [34] | Kadawa, Nigeria | station | Garlic | planting spacing | drip | 425 | 15.3 | 3.6 |
| [49] | Pune, India | station | Garlic | Deficit irrigation | Micro sprinkler | 249 | 7.5 | 3.0 |
| [49] | Pune, India | station | Garlic | Deficit irrigation | Micro sprinkler | 374 | 10.8 | 2.9 |
| [49] | Pune, India | station | Garlic | Deficit irrigation | Micro sprinkler | 498 | 12.9 | 2.6 |

* 10 kg ha$^{-1}$ mm$^{-1}$ = 1.0 kg m$^{-3}$; [1] GH = green house.

## 4. Conclusions

Water use, crop yield, soil moisture, and irrigation water use efficiency of conservation agriculture and conventional tillage were compared for irrigated onion and garlic. Amount of water added was determined either by farmers or based on climatic data. Compared with conventional tillage, in conservation agriculture, irrigation water use is less, the soil moisture content is higher, and crop yield is greater. On the other hand, farmer scheduled irrigation used approximately twice the amount of water than the climate data-based scheduling under conservation agriculture. Onion and garlic yields were approximately 40% greater in conservation agriculture over conventional tillage. The yield of onion from climate data-based scheduling treatment was 63% greater than the farmer's irrigation practice, while it was about 41% higher for garlic vegetable production. Similarly, for onion 44% and for garlic 57% greater irrigation water use efficiency was obtained in conservation agriculture than conventional tillage treatment. In both years, there was lower irrigation water use efficiency under farmers' practice due to low yield of onion and garlic as a result of over-irrigation at initial stages. Due to greatly increased yields and water savings under conservation agriculture in smallholder plots, farmers applied grass mulch and used no-tillage practices. Adoption of conservation agriculture by smallholder farmers during the dry phase has social and economic benefits because less labor was required for tillage and irrigation water application. However, additional research is needed in grass mulch availability and pest occurrence under conservation agriculture if these benefits can be achieved in a large-scale implementation.

**Author Contributions:** S.A.B. has contributed to the conceptualization, methodology, investigation, data collection and acquisition, data analysis, writing the original draft manuscript in scientific content. P.S. contributed to the methodology and editing of the manuscript. A.W.W. contributed to the data collection framework, data analysis and review and editing of the manuscript. T.S.S. contributed to the conceptualization, methodology, methods of data presentation, visualization, supervision and in rewriting the manuscript to the scientific content. M.R.R. contributed to the conceptualization, methodology, design of the experiment and manuscript editing. S.A.T. contributed to the conceptualization, methodology, data collection and acquisition framework, visualization, supervision, project administration, funding acquisition and rewriting the manuscript to the scientific content.

**Funding:** This research was funded by the American people through support by the United States Agency for International Development (USAID) Feed the Future Innovation Lab for Collaborative Research on Sustainable Intensification (Cooperative Agreement No. AID-OAA-L-14-00006, Kansas State University) through Texas A&M University's Sustainably Intensified Production Systems and Nutritional Outcomes, and University of Illinois Urbana-Champaign's Appropriate Scale Mechanization Consortium projects. This work was co-financed through the CGIAR Research Program on Water, Land, and Ecosystems (WLE). The contents are the sole responsibility of the authors and do not necessarily reflect the views of USAID. The APC was also funded by USAID.

**Acknowledgments:** We would like to acknowledge the Ethiopian National Meteorological Agency (ENMA) for providing longer year climate data.

**Conflicts of Interest:** The authors declare no conflict of interest.

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
