# Peer review of "Conservation Agriculture Saves Irrigation Water in the Dry Monsoon Phase in the Ethiopian Highlands"

_water, doi:10.3390/w11102103_

Round 1

Reviewer 1 Report

Dear Authors, 

Firstly the Figure 1 is not making meaningful sense. Kindly revise it.

Secondly, the conclusions drawn from this study requires more field data from different years. Claiming the output just from one year makes the study questionable. Any kind of crop modeling effort supporting your claim may help. 

There are figures, like Fig 3 and some others in the manuscript which are blurry or pixels are distorted, kindly revise them. 

Author Response

Thank you for your valuable comments to improve our manuscript.

Reviewer 2 Report

I have uploaded a pdf with my specific comments on the manuscript. Please let me know if there are any difficulties reading these comments. Most of my suggestions are minor edits. There is one major concern I have regarding Table 4 and the calculation of WUE that needs to be addressed prior to publication.

Author Response

(The authors gave the same response as above.)

Reviewer 3 Report

This paper describes the benefits of conservation agriculture practices for vegetable farmers in Ethiopia.  I believe it is potentially publishable if the authors address the following points AND particularly if they improve the English throughout the manuscript.

Line 54. Reference 14 should not be used, because readers cannot access it. Line 76 – Do they really farm 35% slopes? Please list the slopes of the actual plots that the research was conducted on. Line 116 – Where does the grass mulch come from? Section 3.1 – How was irrigation water applied in each treatment? Sprinkler? Furrow?  Watering can? 180-186 – This is confusing. First, it says that the same amount of water was applied to T1 and T2; then it says that 14% less water was applied to T1 than T2. 261-300 – There is no indication of variability among plots in any of the soil moisture data. Were these only measured in a single plot for each treatment?  How do we know if these differences are significant? Section 3.4 – This section has many English grammatical and usage errors. Please revise. Lines 376-387 – Again, numerous English usage errors. Line 421 – Do you mean “comparable to” rather than “compared to”? Conclusions – The grass mulch has clear water-saving and yield benefits, but how much labor and water is required to produce, harvest and apply it? Will this be a factor for farmers in deciding whether to use this practice? Please discuss.

Author Response

(The authors gave the same response as above.)

Reviewer 4 Report

I made comments throughout the manuscript, which can be seen in red ink. I'll point out a few comments here:

Please double check dates in Table 3, and it was also not clear if weeding/hoeing was only done in the conventional tillage treatment.

Also, if irrigation was done manually with buckets, how did you know the amount of water that was applied. Please add a sentence to explain this.

Fig. 2 - Is the depth averaged over garlic and onion? Please clarify.

Why different labels on the x-axis in Fig. 3?

Figs. 4 and 5 are difficult to interpret if printed in black and white. Also, I don't understand Fig. 4. You appear to be showing soil moisture before and after irrigation applications for T1 and T2, but how can these ben continuous measurements? 

Most of the third paragraph under SEction 3.4 is redundant and already previously stated.

Check proper use of consistent and inconsistent.

Check y-axis labels on Fig. 6.

Check numbers in first paragraph under Section 3.6 - at least one appears to disagree with number shown in Table 6 (for T3).

The numbers shown for the treatments in this study in Table 7 do not all agree with the values shown in Table 6. Why? Please explain. And why the 4th treatment for garlic. Is it really necessary to show this again in Table 7, even in the numbers do agree?

Author Response

(The authors gave the same response as above.)

Round 2

Reviewer 1 Report

Good Job with the revised version. I would still emphasise the authors to revise Figure 1, I find it not self interpretable. What's the purpose of units on the boundary? Why can't be there a boundary of subwatershed while displaying treatment positions spatially. 

Figure 4 seems to be a screen shot, I did not find any issues with its caption but I believe it can be further improved. 

Other than that, good work!

Author Response

Thank you for your valuable comments to improve our manuscript.

This manuscript is a resubmission of an earlier submission. The following is a list of the peer review reports and author responses from that submission.

Round 1

Reviewer 1 Report

The paper is related to very intersting subject of water scarcity in Sub – Saharan region in Africa and application of environmetaly friendly method as Conservation Agriculture (CA).

The paper is worth to publish because the data can be usful for other growers in this area or Extention Service. There are also interestin for other researchers working with plant irrigation relations.

The obtained results (yield of onion and garlic) in two cultivation methods (CA) and conventional tillage (CT) and  irrigation  management based on meteorogical data are very promising. However,  the correct of  irrigation, accordingly to evapotranspiration (ET) is not easy for common  application.

Most of the meteorological stations give the possibility to calculate ETo  and thus knowing the crop coefficient Kc is possible to calculate the real crop requirements accorging to growth phase.

The paper presents interesting data, is well organized with many detailed data related to water content in soil, soil physical parameter, climatic data and informations related to crop managements.  Intersting data are also presented in Table 7 with data from literature related to onion and garlic yields at different irrigation (applied water levels). Paper requires some improvements and cotrrections. The detailed comments are given in the text body. Especially the calculation of IWUE needs correction  Calculation of IWUE given in table 7 (from literature) are correctly presented. However, in  the Table 6 results from experiment requires some correction. I am not sure if the yield (from how big area or several field? is presented) or producticivity is well presented). Please check again the results and calculation of IWUE. The discussion is related to the obtained results but first they should be realiable.

Reviewer 2 Report

I must appreciate the authors for the level of efforts they had to put in making this study happen. I want to suggest few changes:

There is no climate variable analysis, no time series analysis of Precipitation or temperature (observed data)

All the figures and tables need extensive formatting based on journal guidelines. All Figures are distorted and Figure 1 & 2 needs remake. Figure 3 has distorted labels. Figure 5 is mismatched in height.

The feasibility in terms of social and economic constraints should be briefly addressed in conclusions. 

Discussion and Result section should be made one and very few reviews (literature) are discussed in the text. Add more comparative study if not from study region but similar agro-climatic zone. 

Introduction can be expanded while emphasizing and explaining CA in more detail. 

The journal guidelines are not followed in most part of the manuscript. 

There could be better way of explaining the data with control. 

Reviewer 3 Report

Dear authors,

I have carefully revised your manuscript and there are many reasons to reject this manuscript in it'd present form.

English language is not adequate. 

Treatments are not properly described. For instance, can you explain the difference between both irrigation strategies? According to your explanation, it seems that one of them consisted in irrigation scheduling based on climate data, but, and the other one?

Other question would be related to the text organization. For instance; soil properties are shown in results, but this information should be included in M&M.

In the section 3.2, the whole of data do not correspond to the information provided in Table 4.

During the Discussion section, you continue showing Results, and all these figures should be moved to Results section.

Quality figures is not good. 

Finally, relating to the References, the number is appropriate, even larger than it is necessary. 

However, if you revise them, only 22 references correspond to the last ten years. From my point of view, you could select the most recent of them and simplify, because, on overall, the discussion is not as relevant to include this number of references.

Finally, I consider that if this manuscript is reorganized and re-written, it could be re-considered to be published.